# Hypertension management: experiences, wishes and concerns among older people—a qualitative study

Emma van Bussel,[1] Leony Reurich,[1] Jeannette Pols,[1] Edo Richard,[2,3] Eric Moll van Charante,[1] Suzanne Ligthart[4]

[1]Department of Primary Care and Medical Ethics, Amsterdam UMC, AMC, Amsterdam, Netherlands
[2]Department of Neurology, Amsterdam UMC, AMC, Amsterdam, Netherlands
[3]Department of Neurology, Radbout University Medical Center, Nijmegen, Netherlands
[4]Department of Primary and Community Care, Radbout University Medical Center, Nijmegen, Netherlands

**Correspondence to**
Dr Eric Moll van Charante;
e.p.mollvancharante@amc.uva.nl

## ABSTRACT

**Objectives** Sixty-five per cent of older people have hypertension, but little is known about their preferences and concerns regarding hypertension management. Guidelines on hypertension lack consensus on how to treat older people without previous cardiovascular disease (CVD). This asks for explicit consideration of patient preferences in decision making. Therefore, the aim of this study was to explore older peoples' experiences, preferences, concerns and perceived involvement regarding hypertension management.

**Design** Qualitative interview study.

**Setting** Participants were selected from 11 general practitioner (GP) practices in the Netherlands and purposively sampled until data saturation was achieved. Semistructured interviews were conducted, audio recorded and analysed by two researchers using thematic analysis.

**Participants** Fifteen community dwelling older people aged 74–93 years with hypertension and without previous CVD participated.

**Results** Interviewees rarely started the conversation about hypertension management with their GP, although they did have concerns. Reasons for not discussing the subject included low priority of hypertension concerns, reliance on GPs or trust in GPs to make the right decision on their behalf. Also, interviewees anticipated regret of reducing medication, fearing vascular incidents. Interviewees would like to discuss tailoring treatment to their needs, deprescription of medication and ways to reduce side effects. They expected GPs to be more transparent on treatment effects.

**Conclusion** Older people describe having little involvement in hypertension management, although they have several concerns. Since GPs are also known to be hesitant to bring up this subject, we signal a conspiracy of silence about antihypertensive medication. Through breaking this silence, GPs can facilitate shared decision-making on hypertension management and better tailored care.

## INTRODUCTION

General practitioners (GPs) consider hypertension management for older people challenging.[1] For people aged 70 years and over, primary prevention guidelines on cardiovascular disease (CVD) lack consensus on (de) prescribing antihypertensive medication (AHM). Treatment decisions are further complicated by multimorbidity, shorter life-expectancy, functional decline and attenuating benefit-to-harm ratios.[2–5] These uncertainties underline the need for patient involvement in decision making, to tailor care.[5] GPs wish to take into account patients' preferences and circumstances. However, in everyday practice, these are not always optimally addressed and are generally not discussed.[1 6 7]

Studies on lay perspectives on blood pressure have shown a gap between patients' and doctors' understanding of hypertension and the need for AHM, calling for acknowledgement of these differences and engagement of patients.[8] At present, little is known about older patients' involvement in and preferences on hypertension management in primary care. This is surprising since 65% of older people have hypertension and 82% of them are using AHM.[9] Previous studies showed that older people wish to be involved in decision-making about their medical conditions, and expect their GP to consider their personal preferences, situation and concerns in decision-making.[10 11] However, GPs often do not appear to apply shared decision-making (SDM) in consultations with older patients.[12]

Insight into older patients' experiences regarding hypertension management can support GPs in involving patients in decision

### Strengths and limitations of this study

► Interviews were performed at the participants' homes, enabling them to speak freely about their experiences, wishes and concerns.
► A diverse population of older persons was interviewed.
► Few participants experienced side effects of their medication.
► Information on decision making at the very start of hypertension treatment was limited, due to recall bias.

making and thus in tailoring hypertension management. With this study, we aimed to explore older peoples' experiences with AHM, how they perceive their involvement in hypertension management and what their preferences and concerns regarding hypertension management are.

## METHODS
### Participants
For this qualitative interview study, community dwelling hypertensive persons aged 70 years or older were recruited from GP practices. We included participants without a history of CVD because people who experienced CVD may have different motivations for preventing future CVD and taking preventative medication, compared with people without CVD. To ensure diversity, GPs were asked to purposively select older persons based on age, gender, educational level, geographical region, urbanisation, functional abilities and experiences with adverse effects of AHM. GPs recruited a maximum of two patients per practice. During the iterative process of interviewing and analysing, new GPs were asked to recruit patients with specific characteristics that were, so far, underrepresented in our sample. All participants who agreed to have their contact details passed

on to the researchers signed written informed consent and participated. A gift voucher (€10) was offered to participants as compensation for their time.

### Data collection
Semistructured interviews with use of a topic list were held at the participants' homes by LR, EB or both between January and March 2018. Interviews were conducted in Dutch language. In 2018, LR and EB were both female GP trainees who had worked a year in a GP practice and with experience in qualitative and quantitative research. To avoid participants from feeling hesitant to report criticism about GPs, the interviewers introduced themselves as researchers and emphasised that results were confidential and would not be disclosed to the participants' GP. The interview started after introductions, small talk and reiterating the research topic and goal. All participants were interviewed on their views on hypertension, reasons to (not) treat it, the experienced and desired role of the GP and their own role in treatment decision making. The topic list was modified during the study period, as new themes arose during the first interviews. The interview topic list is presented in table 1. Once data saturation was achieved (after 15 interviews), no new participants were

| Table 1 | Interview topic list |
| --- | --- |
| **Topic** | **Subtopic** |
| Participant characteristics | Demographics, including medical history and medication use |
| Hypertension and antihypertensive medication | Frequency of hypertension visits at the GP practice |
| | Topics discussed during hypertension visits at the GP practice: medication use, side effects, lifestyle |
| | Experience with hypertension, antihypertensive medication (use) and lifestyle (changes) since hypertension was first identified |
| | Attitudes towards hypertension, antihypertensive medication (use), lifestyle and necessity of treatment |
| | Experiences with and attitudes towards deprescribing antihypertensive medication |
| Shared decision-making | Received information on hypertension, risk and risk reduction from antihypertensive medication and lifestyle during (hypertension) visits at the GP practice |
| | Experience of being involved in decision making and receiving tailored information and care |
| | Views on involvement and tailoring information and care by GP |
| | Experience of trust in the GP and in hypertension management in primary care |
| | Experiences of GPs' uncertainties in hypertension management in older persons |
| | Experience of and views on necessity, barriers and facilitators to discuss hypertension management with the GP |
| Personal situation and preferences | Experience of and views on consideration of personal circumstances and age in decision making in hypertension management |
| | Experienced effects of one's personal situation on views on hypertension and its management |
| | Experienced prioritisation of facets of hypertension and its management |
| | Experienced effect of chronic hypertension and long-term medication use on views on hypertension and its management |
| Future perspectives | Wish to change things regarding hypertension management: lifestyle, antihypertensive medication use or involvement in decision making with the GP |

GP, general practitioner.

van Bussel E, et al. BMJ Open 2019;9:e030742. doi:10.1136/bmjopen-2019-030742

recruited. Field notes were made after each interview. The duration of the interviews ranged from 20 to 90 min.

## Analysis

All interviews were audio-recorded and transcribed verbatim. Qualitative analysis of the data was performed following the six phases of thematic analysis according to Braun and Clarke:[13]

1. First, two authors (LR and EB) familiarised themselves with the collected data by transcribing the interviews verbatim and repeatedly listening to and reading the data.
2. LR and EB coded the first six interviews independently by systematically going through the data, creating initial codes. After comparing three independently coded interviews, codes were merged. The next three interviews were coded independently and compared, resulting in a new set of codes. The other interviews were coded by one author using these codes and checked by the other. The coding was compared and discussed until agreement was found.
3. Identified codes were sorted into potential themes which were discussed by the authors (LR, EB, JP, EMC, SL) until consensus about potential themes was reached.
4. Potential themes were reviewed, and thematic maps were made. The collected data were re-read to make sure the thematic map was representing the data set. Whether data saturation was reached was discussed.
5. The resulting themes were refined and a narrative of the found data was considered and discussed by LR, EB, JP, EMC, SL.
6. Final analysis was performed, and illustrative examples were selected and translated into English, in order to answer the research question and compare our analysis to existing literature.

Analysts were GP (trainee) (LR, EB, EMC, SL), neurologist (ER) and/or anthropologist (LR, JP). For coding and analysis MAXQDA Plus 12 (V.12.2.1) was used. Reporting of our study is in accordance with the consolidated criteria for reporting qualitative research (COREQ).[14]

### Patient and public involvement

There were no patients involved in the development of the research question, the design recruitment to or conduct of the study. This study was specifically designed to bring together the perceptions of hypertensive older patients, to inform hypertension management in general practice.

## RESULTS

In total, 15 individuals aged 74–93 years (mean 81) from 11 different GP practices were interviewed. Two lived in senior apartments, and all interviewees were living independently. Three received help with groceries and used walking aids. Participants had 3–8 (median 4) prescriptions, with 1–3 (median 2) types of AHM and level of education ranged from primary school to higher education (online supplementary table S1). Partners of three interviewees were present in the same room during the interview and occasionally confirmed, contradicted or supplemented the interviewee. Two interviewees had experienced side effects from AHM.

In the conducted interviews, participants expressed their concerns and preferences regarding hypertension and hypertension management. The results of the interviews were structured in four themes: 'Older peoples' perspectives on hypertension management are not discussed, 'reasons for not discussing needs and preferences regarding hypertension management', 'concerns and preferences regarding hypertension management' and 'uncertainty about implications of potential choices in hypertension management'.

### Older peoples' perspectives on hypertension management are not discussed

Interviewees had regular check-ups for their hypertension management, often with a practice nurse. This manuscript focuses on the GP, because interviewees almost exclusively mentioned the GP as the key professional engaged in their hypertension care. The GP takes decisions on AHM, and participants consider the role of the practice nurse solely for check-up, follow-up and plain advice. In general, participants did not discuss their concerns regarding hypertension management with their GPs or felt barriers to do so. When directly asked why, interviewees expressed that there was no need to discuss the subject or that they experienced barriers to discuss issues on hypertension with their GP. During the interviews however, all participants reported concerns which they did not express to their GP. In general, they expected (and waited for) the GP to start the conversation about their preferences and concerns regarding hypertension management. One participant who felt reluctant taking medication reported feeling extremely relieved when her GP suggested to reduce her AHM.

> 'Then she [the GP] said: "why don't we try to reduce the medication?" At that moment I could hug her, I would never dare to ask this myself (…), but if she wants to!' (P6)

There were four key reasons why older persons did not start the conversation about hypertension management with their GP and four reasons why they desired such a discussion.

### Reasons for not discussing needs and preferences regarding hypertension management

#### Hypertension is not a priority for older people

When participants experience no symptoms of hypertension or side effects of treatment, they generally did not seem to bother much about hypertension or AHM. Interviewees felt hypertension is part of normal ageing, and other diseases or circumstances were more important to them. Taking AHM daily was part of their routine and did not seem to burden these participants.

'I am not bothered by it [hypertension], so I never think about it. I just never think about it' (P7)

## Reliance on the GP

Fear of jeopardising the relation with the GP, on whose care they relied for all other (medical) problems, was a barrier to express concerns regarding AHM.

'You feel dependent on them. You cannot fight with them, because what can you do, it is not easy to change or find another GP (…) so your dependency is bigger than I anticipated.' (P5)

Also, the perceived authority of the GP formed a barrier to discuss any doubts about AHM. Some interviewees felt that it would be inappropriate to question the GP's judgement.

'I don't want to do that [discuss reducing medication], because then I disregard my GP's advice, my GP expects me to follow her advice' (P1)

## The GP knows best

Another reason to remain silent was trust in the GP and his or her expertise. Interviewees were convinced that the GP would continuously monitor their medication and trusted him to start the conversation if this was indicated.

'I just assume that if he thinks I need another pill, that he knows what is best for me' (P2)

The long-term relation with the GP enforced this trust in the GP's expertise.

'I thought, well, the doctor is probably right, who am I, I didn't study for it, so he probably knows, because he knows me already for a long time'. (P2)

Interviewees found it difficult to truly understand the consequences of risk (reduction) for their personal situation and to interpret that information. This was a barrier to making decisions and discussing doubts with their GP. The uncertainty made interviewees rely on the advice of their GP; the expert.

'I would like to know [statistics on risk reduction], but I would like to get an answer which says something about my own risk, but there is no such answer, that makes it difficult.' (P5)

## Anticipated regret and avoiding responsibilisation

Interviewees were aware of the threats of hypertension. They feared CVD, particularly cerebrovascular accidents, since it often comes with disability, loss of independence and loss of social contacts. One of the strongest motivators not to express concerns about drug treatment was anticipated regret of developing CVD, when reducing medication.

'I don't want to have a stroke (…). I have worked as an activity coordinator in nursing homes. I have seen the results of strokes, I don't want to experience that.' (P10)

In addition, actively initiating change in hypertension management felt like taking responsibility for the outcome. Following doctor's advice, and ignoring concerns about hypertension management was a way to avoid active decision making and responsibility for one's own situation.

'I am not going to be stubborn, because if something would happen, it would be my own fault.' (P2)

In the same line, fear of what might happen kept interviewees from wanting to learn about the details and risks of their condition.

'No, I don't want that [more explanation of the doctor about AHM], no. I mean, if he explains to me what will happen if I don't take them [medication] I would feel uneasy, I think.' (P7)

## Concerns and preferences regarding hypertension management

### Wish to reduce antihypertensive medication

Even if they did not discuss this with their GP, when asked the interviewees showed interest in reducing medication and had questions about their use of medication. First, there was a wish to minimise overall medication use. Second, participants wondered whether (some of) the medication could be deprescribed, after a certain period of adequately controlled blood pressure. They were curious to find out if hypertension would reappear after stopping medication.

'My blood pressure is always good, my question is, is that because of the medication? Or maybe I do not need the medication?' (P14)

### Wish to consider one's age and situation

Taking into account one's age and situation when deciding on continuation of AHM was considered important. For instance, a participant expressed the concern that his GP would not (sufficiently) adjust his blood pressure levels to his older age:

'Sometimes I feel that they think you are still a young guy, because you have to reach certain [blood pressure] levels, which might not be relevant anymore for older people.' (P4)

Some participants thought that relatively high levels of stress in the past might have contributed to increased blood pressure levels at the start of their treatment and wondered whether improvements in their well-being could have had beneficial effects, allowing for deprescription of AHM.

'"Your blood pressure is too high", well that was based on certain levels, it is possible, I don't have knowledge about that (…), but I keep thinking that it was a particular moment'. (P14)

### Side effects

Fatigue, fogginess, dizziness, staggering gait, polyuria, nausea, burning sensation in the stomach and reduced

van Bussel E, *et al. BMJ Open* 2019;**9**:e030742. doi:10.1136/bmjopen-2019-030742

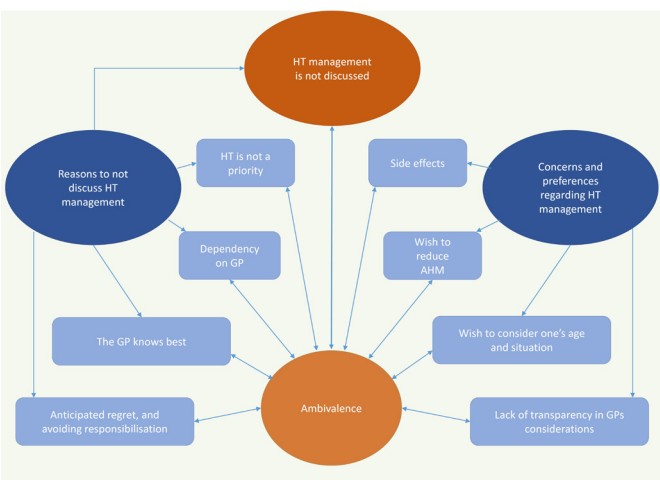

**Figure 1** Final thematic map. Graphical display of the four themes of this qualitative interview study, their subthemes and mutual relations. AHM, antihypertensive medication; GP, general practitioner; HT, hypertension.

exercise tolerance were side effects of AHM that participants reported. These side effects reduced quality of life and resulted in the wish to re-evaluate the benefits and harms of continuing or stopping AHM. For some, this was a reason to discuss hypertension treatment with their GP.

'If it stays like it is at the moment, I think I will accept it. But that burn in your stomach and the feeling that you have to throw up, I don't like that, that was not acceptable for me. It needs to be in my opinion within reason, these side effects. If it becomes too much, I won't accept it. I would then say: "I don't want that drug anymore".' (P15)

### Lack of transparency in GPs' considerations
The interviewees who experienced side effects of AHM learnt about the uncertainty of their GP regarding hypertension management, during their struggle to find a balance between prevention of CVD and side effects. Despite the knowledge that GPs do not know the optimal blood pressure targets in older age and thus the right treatment, one of them felt reluctant to discuss his own doubts with the GP. Still, he would have appreciated it if his GP had shown and shared more of his uncertainty, to make room for alternative outcomes (instead of blood pressure target as the only goal).

'They really don't know. They have statistics, but whether you meet those or not, they don't know, they just don't know. I think they should be more transparent about that.' (P5)

### Uncertainty about implications of potential choices
Overarching the aforementioned themes is the uncertainty of implications of choices regarding hypertension management. Interviewees were ambivalent about deciding which of their concerns and preferences were most important and should be acted on.

'It [taking antihypertensive medication] is a compromise. (…) Of course, if it would be better [to take more medication] and if it lowers my risk I would accept it. I wouldn't be happy about it. I would be reluctant [to take medication] but it would be necessary, it would be better because you would lower your risks.' (P3)

The final thematic map is shown in figure 1.

## DISCUSSION
Older people do not feel sufficiently involved in decision making about hypertension management in general practice, while they do have many unexpressed concerns. Reasons not to raise these concerns are the low priority compared with other complaints or diseases, being reliant on their GP, trust in their GP to make the right decision on their behalf and anticipated regret of incident CVD in absence of treatment. Participants could benefit from overcoming these barriers but wait for their GP to bring up the subject. They often have the wish to reduce medication, to tailor management to their individual situation, to avoid side effects and to receive transparent information on expected treatment effects. Last, participants experienced ambivalence regarding their concerns and preferences in hypertension management, hampering their perceived capacity for decision making on their own situation.

Strengths of this study include that interviews were performed at the participants' homes, by researchers who were independent of their GPs. This way, we enhanced trust and enabled a diverse population of older persons to speak freely about their experiences, wishes and concerns. Because of the background of the authors (GPs and anthropologists), the analyses had a pragmatic but open character, looking for themes that may inform and improve primary care. The population was diverse in terms of age, sex, comorbidities, educational level and geographical region. In terms of dependency and medication use, the population was less diverse, with all participants being independent to some extent and taking multiple medications. With only two participants (recently) experiencing side effects of AHM, this groups representation was small in our study, which may have limited our findings. It was difficult to include persons with side effects from AHM, since most older persons were using AHM for a long time without any problems, with difficulty to recall past episodes of problems or reluctance to discuss this with their GP. It is reported that in adults up to 70 years of age, 85% of those starting AHM experienced side effects,[15] while of long-term AHM users (up to 84 years), 20% reported side effects.[16]

Although we aimed to include people without a history of CVD, two participants probably had had CVD (P9 and P15) judging from the interviews and their medication list (acenocoumarol, acetylsalicylic acid). When selecting participants, we relied on the GP to have assessed the inclusion criteria. However, in our study opinions did not

differ from participants with definite absence of CVD in their medical history, particularly not on motivation for preventing future CVD or willingness to use preventative medication. Decision making at the very start of hypertension treatment was hard to recall, limiting the information on treatment initiation. Yet, a large proportion of Dutch older adults have long-term hypertension and our results apply to this group.[9]

Our results provide insight into experiences and views of older persons with hypertension, that may apply to other developed countries. Although the next generation of older people may have more difficulty with accepting authority in GPs and less problems with speaking up about their concerns, the complexity of decisions in hypertension management is not likely to change.

## A conspiracy of silence

There appears to be a conspiracy of silence, in which both the older patient and the GP do not express their concerns regarding antihypertensive treatment. From previous studies, it is known that GPs feel that they hardly involve older persons in decisions on hypertension management.[1 6 7] Reasons are time constraints, automated prescriptions, negative emotional impact on patients and anticipated regret in case of cardiovascular events.[6] This study showed that older people too have the impression of no or minimal involvement in their hypertension management in primary care. From prior studies, little is known about (the perceived) involvement of older people in their hypertension management.[17] This contrasts the wish for greater patient involvement in hypertension management among both GPs and older patients.[1 6 7]

## Reasons to remain silent

There is uncertainty and ambivalence in prioritising preferences and concerns regarding AHM use among older people in our study. Ambivalence has previously been shown in qualitative studies among older people. AHM can provide a feeling of security regarding prevention of CVD, but older patients also worry about side effects and long-term adverse effects of taking chronic medication,[18–20] are cautious to take preventative medication and prefer minimising medication-use.[19 21] Concerns are generally outweighed by the perceived necessity of treatment and their GP's recommendation.[21 22] In our study, side effects were the only reason to discuss alternatives with the GP.

Reliance on the GP was a major barrier that is also recognised in SDM in other fields. Patients are in a vulnerable position and can feel too dependent to bring up concerns in interaction with the doctor.[23] Other barriers to active involvement in decisions were reluctance to bear responsibility for cardiovascular outcomes and anticipated regret, also known from SDM in other fields.[23] It was previously shown that anticipated regret has stronger associations with health behaviour than other anticipated negative emotions and risk appraisals.[24]

In our study, hypertension and hypertension management were generally given low priority. When taking AHM, participants felt safe in terms of preventing CVD morbidity and mortality. However, older people overestimate both the risks of hypertension and the benefits of medication. For example, hypertensive people with a mean age of 73 years estimated their stroke risk to be 40% within 5 years and expected over 50% stroke risk reduction from AHM, while in fact 5-year risk of cardiovascular events for this group is 5%–20% and stroke risk reduction from AHM is 24%–42%.[25–27] After being shown the actual risks and benefits, a quarter of 75 older patients with hypertension participating in a qualitative study became uncertain about AHM use or would decline to take it.[25] Trust in GPs and AHM seems high and also unrealistic, so participants' choices are often based on false or at least incomplete information. Concerns about taking medication, side effects and tailoring to the individual situation could receive different prioritisation if patients would be better informed and if GPs would be more transparent on their uncertainty.

## Breaking the silence

It is important to break the silence since older patients generally wish to be involved in decision-making about their medical issues, despite the abovementioned reasons to remain silent.[10] GPs also value patient involvement in decision making, especially since guidelines leave room for discussion.[6 28] By breaking the silence, GPs could guide older patients in their ambivalent feelings about AHM and improve personalised care. For example, once started, lifelong use of AHM is the current standard of care. However, a systematic review showed that after 2 years of withdrawal of AHM, hypertension did not return in 26% of patients (mean age 41–76 years). This justifies an AHM withdrawal attempt in well controlled patients, if they wish to do so.[29] In addition, more transparency could reduce misinterpretations and lower the threshold to start discussing factors important to patients in hypertension management.

Heterogeneity exists in how older people want to be involved in decision-making. Older patients with multimorbidity and healthcare professionals agreed that SDM in older patients requires a continuous dialogue between professional and patient.[30] Contextual factors for receiving information, such as having enough time and having a good relationship with professionals involved, are considered of great importance.[31] Elements in decision making unique to older people are the opportunity for input from trusted others and discussion of decisions' impacts on patients daily lives.[32]

Older persons generally do not start the dialogue themselves; they wait for their GP to do so. In our study and also in SDM in other fields of medicine, older patients experience barriers to participate in decision making about their care.[33] Since most barriers are related to the GP, including the authority ascribed to them as well as time constraints, we recommend that GPs take the initiative to break the silence.[22]

## Implication for practice

To break the conspiracy of silence, GPs should explicitly and repeatedly discuss hypertension treatment with their patients on chronic AHM. This is the only way to come to true SDM about hypertension treatment and may lead to adaptation and potentially cessation of AHM in those for whom the benefits not clearly outweigh the burden.

**Acknowledgements**  We sincerely thank all individuals who participated in this qualitative study and all general practitioners and their assistants who selflessly recruited participants.

**Author contributions**  EvB, LR, JP, ER, EMvC and SL contributed to the conception or design of the work. EvB and LR performed the interviews. EvB, LR, JP, ER, EMvC and SL contributed to the acquisition, analysis or interpretation of data for the work. EvB drafted the manuscript. EvB, LR, JP, ER, EMvC and SL critically revised the manuscript and gave final approval and agree to be accountable for all aspects of work ensuring integrity and accuracy.

**Funding**  This research was supported by the Netherlands Organisation for Health Research and Development (grant number 839110003).

**Disclaimer**  The funders had no role in study design, methods, subject recruitment, data collection and analysis, interpretation of data, preparation of the manuscript or the decision to publish. No financial disclosures were reported by the authors of this manuscript.

**Competing interests**  None declared.

**Patient consent for publication**  Not required.

**Ethical approval**  Official approval of this study was waived by the Medical Ethics Review Committee of the Academic Medical Centre in Amsterdam (reference W18_004#18.015), the Netherlands.

**Provenance and peer review**  Not commissioned; externally peer reviewed.

**Data availability statement**  Data underlying our findings cannot be made publicly available for ethical reasons; public availability would compromise our participants' privacy. Data requests may be sent to the corresponding author at e.p. mollvancharante@amc.uva.nl.

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
