## [Reviewer comments · BMJ Open]

ARTICLE DETAILS

TITLE (PROVISIONAL)	Hypertension management: experiences, wishes and concerns among older people – a qualitative study
AUTHORS	van Bussel, Emma; Reurich, Leony; Pols, A.; Richard, Edo; Moll van Charante, Eric; Ligthart, Suzanne

VERSION 1 - REVIEW

REVIEWER	Iain Marshall King's College London, UK
REVIEW RETURNED	24-May-2019

GENERAL COMMENTS	This paper describes a qualitative study conducted in the Netherlands, examining the perspectives and experiences of older people with hypertension. The study included 15 participants, which is on the small side but acceptable for a qualitative study. The authors describe that they recruited until data saturation was reached. The study used semi-structured interviews, which were transcribed, and a thematic analysis conducted. Overall, the study appears to have been well done, but I feel it would benefit from some more detail in the reporting in several areas, and also would be improved by setting the results better in context of the (large) qualitative literature in hypertension which currently isn't much mentioned. Methods section: some more detail would be helpful particularly around the recruitment. 1. The authors describe purposive sampling, based on age, gender, education, geographical region, urbanisation, functional abilities, and experiences with adverse effects. However, this is more of an aim than a specific method. How did the researchers enact this? How did the researchers have advance knowledge of these characteristics of the participants they approached somehow? In particular, I think this would be challenging given that participants were approached by their own GP (and participating GPs in practice each recruited 1 or 2 participants). On the face of it, it sounds implausible that the recruitment could be done this way, but the authors could clarify exactly what was done. Perhaps it is to do with terminology - I think there is a subtle difference between purposive sampling, meaning a deliberate selection of participants aiming for diversity, versus some form of
---

pragmatic sampling (i.e. whomever the GPs happen to have chosen), but with the participants turning out to be sufficiently diverse anyway.

2. The participants are described as 'diverse', though from the table they appear to be diverse in some ways and not others. I think it's probably fair to say they are diverse in terms of age, sex, comorbidities, and educational level. However, they are all highly independent (either completely or with ADLs at least), none are described as rural, and all take multiple medications. In particular, the high level of independence is probably not typical of the older population as a whole, and one limitation is the lack of older people who have more functional limitations.

That the study was done in participants own homes is indeed a strength, particularly given the subject being discussed. The authors could describe if the interviewers introduced as being GPs themselves, and could consider the effect of this on the interview.

Notably on the COREQ checklist the authors provide, there are relevant unfilled points - 'participant knowledge of the interviewer', and 'interviewer characteristics'. That the interviewers were GPs, and also that the thematic analysis was conducted by GPs might have influenced their approach to the data, and is worthy of mention.

p4 l18 'Little is known about older patients...'

There is a large body of qualitative research in older patients with hypertension which is not currently mentioned. For example, our own group conducted a review in 2012 (<https://www.bmj.com/content/345/bmj.e3953>) which found 53 qualitative studies looking at patient perspectives on hypertension, and from all parts of the world. A number of these were in older people. I know that a number of similar studies have been published subsequently. The authors should consider setting their study better in the context of the very large number of other relevant studies done before on the topic, both in the background and discussion.

p6 l17 'Hypertension management is not discussed': perhaps more accurately that the patients' concerns were not discussed?

p6 l19 'often with a practice nurse' - despite this important point that most of the hypertension care is from a nurse, there seems to be a strong focus on GPs throughout. Did participants describe their experience seeing the nurse? And did they feel more comfortable discussing their medication concerns with their nurse?

p6 l24 'unexpressed concerns' - perhaps better 'concerns which they did not express to their GP'

p6 l40 grammar not quite right, suggest reword

p6 l42 Hypertension is not a priority for whom?

p5 l55 'Dependency' - this is often used to mean that a doctor perceives a patient is consulting too frequently and inappropriately, which I don't think is the meaning intended here. Perhaps clearer to put something like 'Concern about harming relationship with doctor' as the theme header?

	Minor corrections p2 l26 'Interviewees rarely started the conversation' - as written reads to me as if describing the conversation with the researchers. Perhaps say explicitly that with their GP. p2 l36 'Older people have little involvement...' - perhaps a change to wording to 'Older people describe having little involvement...' p4 l10 use of 'AHM' acronym - I don't believe this is a standard acronym, (although it is defined) - perhaps the editor could advise on journal style. p7 l45 'responsibilisation' - not a word I've seen Table S1 caption, c - I found this sentence difficult to parse, please consider rewording Were interviews were conducted in Dutch language, and translated by the authors for this paper? If so, this could be mentioned.
--	---

REVIEWER	Janet Hanley Edinburgh Napier University UK
REVIEW RETURNED	18-Jun-2019

GENERAL COMMENTS	This paper is remarkably succinct and very clear. However the reader may be helped by a little more detail in one or two places. There is a statement on the overall design in the abstract, but not in the main paper. It would help if that was included at the start of the methods section. Similarly it would help if immediately prior to the presentation of themes and subthemes there was a sentence introducing what these headings are. It is not clear how the purposive sampling worked when GPs were asked to recruit patients - were they given a list of the variables? A little more detail is required here. There is no reflexive account of how the background of the interviewers/ analysts may have influenced the findings. There is also no account of what the interviewers told the interviewees about their background. This is marked as no applicable in the COREQ statement, but the interviewers must have introduced themselves in some way. This should be added to the paper and COREQ statement
--

VERSION 1 – AUTHOR RESPONSE

Reviewer(s)' Comments to Author:

Reviewer: 1

Reviewer Name: Iain Marshall

Institution and Country: King's College London, UK

Please state any competing interests or state 'None declared': None

Please leave your comments for the authors below

This paper describes a qualitative study conducted in the Netherlands, examining the perspectives and experiences of older people with hypertension.

The study included 15 participants, which is on the small side but acceptable for a qualitative study. The authors describe that they recruited until data saturation was reached.

The study used semi-structured interviews, which were transcribed, and a thematic analysis conducted.

Overall, the study appears to have been well done, but I feel it would benefit from some more detail in the reporting in several areas, and also would be improved by setting the results better in context of the (large) qualitative literature in hypertension which currently isn't much mentioned.

We thank the reviewer for his critical review of our work, and the constructive comments. We have added information in several sections, as described in detail below. We feel the manuscript has improved substantially with these changes.

Methods section: some more detail would be helpful particularly around the recruitment.

1. The authors describe purposive sampling, based on age, gender, education, geographical region, urbanisation, functional abilities, and experiences with adverse effects. However, this is more of an aim than a specific method. How did the researchers enact this? How did the researchers have advance knowledge of these characteristics of the participants they approached somehow? In particular, I think this would be challenging given that participants were approached by their own GP (and participating GPs in practice each recruited 1 or 2 participants).

On the face of it, it sounds implausible that the recruitment could be done this way, but the authors could clarify exactly what was done.

Perhaps it is to do with terminology - I think there is a subtle difference between purposive sampling, meaning a deliberate selection of participants aiming for diversity, versus some form of pragmatic sampling (i.e. whomever the GPs happen to have chosen), but with the participants turning out to be sufficiently diverse anyway.

We thank both reviewers for commenting on this point, we agree the manuscript would benefit from providing more detail on this topic. The process of sampling is now described as follows:

'To ensure diversity, GPs were asked to purposively select older persons based on age, gender, educational level, geographical region, urbanization, functional abilities and experiences with adverse

effects of antihypertensive medication. GPs recruited a maximum of two patients per practice. During the iterative process of interviewing and analysing, new GPs were asked to recruit patients with specific characteristics, that were, so far, underrepresented in our sample.'[line 99-104]

2. The participants are described as 'diverse', though from the table they appear to be diverse in some ways and not others. I think it's probably fair to say they are diverse in terms of age, sex, comorbidities, and educational level. However, they are all highly independent (either completely or with ADLs at least), none are described as rural, and all take multiple medications. In particular, the high level of independence is probably not typical of the older population as a whole, and one limitation is the lack of older people who have more functional limitations.

The diversity of the population may not be totally clear from the submitted manuscript. Regarding geographical region we feel we covered both rural and non-rural areas of the Netherlands, by including participants in big cities, small cities, towns and villages. Thanks to this comment we realise that we may have used the inappropriate English term (town) to describe villages. We checked the number of inhabitants per 'town', and now distinguish 4 forms:

- City (>100,000 inhabitants)
- Small city (10,000-100,000 inhabitants)
- town (5,000-10,000 inhabitants) and
- village (< 5,000 inhabitants)

Changes were made to Table S1, accordingly, and explained in the legend.

We hope that with the clarification on 'dependency' in Table S1 (also see our response to "minor corrections: Table S1 caption, c"), we made it clear that we included both older people who are living independently, and older people who depend on others for doing groceries and/or doing household chores. In the discussion section the sentence was amended:

The population was divers in terms of age, sex, comorbidities, educational level and geographical region. In terms of dependency and medication use, the population was less divers, with all participants being independent to some extent and taking multiple medications. [line 318-321]

3. That the study was done in participants own homes is indeed a strength, particularly given the subject being discussed. The authors could describe if the interviewers introduced as being GPs themselves, and could consider the effect of this on the interview.

The interviewers were both GP trainees, and researchers. To avoid reluctance to report criticism about their GP and primary care in general, the interviewers introduced themselves as researchers, and emphasised that they were independent of the participants' GP, and that results would not be reported to the participants' GP. The following was added to the methods section: 'To avoid that participants would feel hesitant to report criticism about GPs, interviewers introduced themselves as researchers and emphasized that results were confidential and would not be disclosed to the participants' GP.' [line112-114]

Also see our response to comment 2 of Reviewer 2, where we describe the changes we made to describe the influence of the background of the analysts on the analyses.

Notably on the COREQ checklist the authors provide, there are relevant unfilled points - 'participant knowledge of the interviewer', and 'interviewer characteristics'. That the interviewers were GPs, and also that the thematic analysis was conducted by GPs might have influenced their approach to the data, and is worthy of mention.

This information was added to the manuscript (as described above), and references to the respective page numbers were added to the COREQ checklist.

p4 l18 'Little is known about older patients...'

There is a large body of qualitative research in older patients with hypertension which is not currently mentioned. For example, our own group conducted a review in 2012 (<https://www.bmj.com/content/345/bmj.e3953>) which found 53 qualitative studies looking at patient perspectives on hypertension, and from all parts of the world. A number of these were in older people. I know that a number of similar studies have been published subsequently. The authors should consider setting their study better in the context of the very large number of other relevant studies done before on the topic, both in the background and discussion.

We agree with the reviewer that there is a large body of qualitative research regarding patients' views on hypertension and incorporated additional references on this topic. However research that focuses on older patients' concerns and preferences regarding management is quite scarce and has mainly addressed participants' views on specific (study) programs for cardiovascular risk management.(1) In our study, we openly discussed views, concerns and preferences with older people. Preferences and management in older people may be different from those of middle-aged adults, because in middle-aged adults more evidence exists on how to optimally treat risk factors. In older adults, guidelines lack consensus on (de)prescribing antihypertensive medication, and preferences may shift and become more diverse because of declined life expectancy, comorbidities and functional decline.

To the introduction section of the manuscript we added: 'Studies on lay perspectives on blood pressure have shown a gap between patients' and doctors' understanding of hypertension and the need for antihypertensive medication, calling for acknowledgement of these differences and engagement of patients.(2) At present, little is known about (...)' [line 77-79] and 'However, GPs less often appear to apply shared decision-making in consultations with older patients.(3)' [line 85-86]

Although we agree that studies on older peoples' understanding of hypertension or reasons for non-adherence to antihypertensive medication certainly relate to the subject, we feel that they are beyond the scope of this article.

p6 l17 'Hypertension management is not discussed': perhaps more accurately that the patients' concerns were not discussed?

We changed this heading into: 'Older peoples' perspectives on hypertension management are not discussed' [line 172]

p6 l19 'often with a practice nurse' - despite this important point that most of the hypertension care is from a nurse, there seems to be a strong focus on GPs throughout. Did participants describe their experience seeing the nurse? And did they feel more comfortable discussing their medication concerns with their nurse?

In our manuscript we focused on the GP, because interviewees almost exclusively mentioned the GP as the key professional engaged in their hypertension care, during the interviews. The GP takes decisions on antihypertensive medication, and participants consider the role of the practice nurse solely for check-up, follow-up and plain advice. To clarify the focus on the GP in our manuscript, the following was added to the results section:

'This manuscript focuses on the GP, because interviewees almost exclusively mentioned the GP as the key professional engaged in their hypertension care. The GP takes decisions on antihypertensive medication, and participants consider the role of the practice nurse solely for check-up, follow-up and plain advice.' [line 174-177]

p6 l24 'unexpressed concerns' - perhaps better 'concerns which they did not express to their GP'

The suggested change was adopted in the manuscript. [line 181-82]

p6 l40 grammar not quite right, suggest reword

The heading was changed into: Reasons for not discussing needs and preferences regarding hypertension management. [line 191]

p6 l42 Hypertension is not a priority for whom?

This heading was changed into 'Hypertension is not a priority for older people' [line 192]

p5 l55 'Dependency' - this is often used to mean that a doctor perceives a patient is consulting too frequently and inappropriately, which I don't think is the meaning intended here. Perhaps clearer to put something like 'Concern about harming relationship with doctor' as the theme header?

The word dependency was changed into 'reliance' throughout the manuscript [line 200, 366, 31]

Minor corrections

p2 l26 'Interviewees rarely started the conversation' - as written reads to me as if describing the conversation with the researchers. Perhaps say explicitly that with their GP.

'with their GP' was added to this sentence to make it more explicit. [line 31]

p2 l36 'Older people have little involvement...' - perhaps a change to wording to 'Older people describe having little involvement...'

The suggested correction was made to the abstract. [line 38]

p4 l10 use of 'AHM' acronym - I don't believe this is a standard acronym, (although it is defined) - perhaps the editor could advise on journal style.

The acronym 'AHM' has been used before in the BMJ open,(4) if the editors prefer, 'antihypertensive medication' can be used throughout the manuscript, instead of the acronym.

p7 l45 'responsibilisation' - not a word I've seen

This word has been used in this context before.(5) We feel it covers the phenomenon that we try to describe: the transfer of responsibility (from the GP) to the patient. If the editors prefer, this

phenomenon can be reported in a more descriptive way (e.g. 'Anticipated regret and avoiding to take responsibility').

Table S1 caption, c - I found this sentence difficult to parse, please consider rewording

For language clarity, and a better understanding in the differences of dependency of the participants, this sentence is changed into: 'Independent: able to take care of ADL and doing household chores/ run errands. ADL independent: able to take care of ADL, but dependent on doing household chores/ run errands.' [legend Table S1]

Were interviews were conducted in Dutch language, and translated by the authors for this paper? If so, this could be mentioned.

Interviews were conducted in Dutch indeed, and quotes were translated to English for the purpose of this manuscript. This is added to the methods section: 'Interviews were conducted in Dutch language.' [line 110] and 'and translated into English' [line 140-141]

Reviewer: 2

Reviewer Name: Janet Hanley

Institution and Country: Edinburgh Napier University, UK

Please state any competing interests or state 'None declared': None declared

Please leave your comments for the authors below

This paper is remarkably succinct and very clear. However the reader may be helped by a little more detail in one or two places.

We thank this reviewer for this compliment, the critical review and the suggested revisions that improved the manuscript.

There is a statement on the overall design in the abstract, but not in the main paper. It would help if that was included at the start of the methods section. Similarly it would help if immediately prior to the presentation of themes and subthemes there was a sentence introducing what these headings are.

At the start of the methods section we added: 'For this qualitative interview study (...)' [line 96]

Also, a paragraph to introduce the themes was added: 'In the conducted interviews participants expressed their concerns and preferences regarding hypertension and hypertension management. The results of the interviews were structured in four themes: 'hypertension management is not discussed', 'reasons to remain silent regarding needs and preferences in hypertension management', 'concerns and preferences regarding hypertension management', and 'uncertainty about implications of potential choices in hypertension management'.' [line 165-170]

It is not clear how the purposive sampling worked when GPs were asked to recruit patients - were they given a list of the variables? A little more detail is required here.

This comment is in line with the first comment of Reviewer 1. We agree that more detail is required and specified our selection process in the manuscript. [see response to first comment of Reviewer 1]

There is no reflexive account of how the background of the interviewers/ analysts may have influenced the findings. There is also no account of what the interviewers told the interviewees about their background. This is marked as not applicable in the COREQ statement, but the interviewers must have introduced themselves in some way. This should be added to the paper and COREQ statement

We thank this reviewer for this comment and added the information to the manuscript and COREQ checklist.

How the interviewers introduced themselves, and how this influenced the findings was elaborated on in more detail (see our response to comment 3 of Reviewer 1)

More information on the background of the analysts was added to the methods section: 'Analysts were GP (trainee) (LR, EB, EMC, SL), neurologist (ER), and/or anthropologist (LR, JP).' [line 144]

How the background of the analysts may have influenced the findings was added to the discussion section: 'Because of the background of the authors (GP's and anthropologists), the analyses had a pragmatic but open character, looking for themes that may inform and improve primary care.' [line 316-318]